# MultiMisD: Multimodal Misclassification Detection

## Abstract

The deployment of multimodal models in safety-critical applications, such as autonomous driving and medical diagnostics, requires more than high predictive accuracy; it also demands reliable mechanisms for detecting failures. In this work, we address the largely unexplored problem of misclassification detection in multimodal settings. We propose MultiMisD, a novel framework specifically designed to detect such multimodal failures. Our approach is driven by a key observation: in most misclassification cases, the confidence of the multimodal prediction is significantly lower than that of at least one unimodal branch, a phenomenon we term *confidence degradation*. To mitigate this, we introduce an *Adaptive Confidence Loss* that penalizes such degradations during training. In addition, we propose *Multimodal Feature Swapping*, a novel outlier synthesis technique that generates challenging, failure-aware training examples. By training with these synthetic failures, MultiMisD learns to more effectively recognize and reject uncertain predictions, thereby improving overall reliability. Extensive experiments across four datasets, three modalities, and multiple evaluation settings demonstrate that MultiMisD achieves consistent and robust gains. The source code will be publicly released.

## 1 Introduction

Multimodal models are increasingly adopted in safety-critical domains such as autonomous driving and medical diagnostics. By integrating complementary cues from diverse modalities (e.g., video, audio), they often achieve superior robustness and generalization over unimodal approaches (Feng et al., 2020; Wang et al., 2018). However, even state-of-the-art models can be dangerously overconfident in their erroneous predictions (Zeng et al., 2025), posing serious risks in high-stakes applications. In such settings, detecting untrustworthy predictions is as crucial as achieving high overall accuracy. While prior work in uncertainty estimation (Lakshminarayanan et al., 2017), calibration (Guo et al., 2017), and out-of-distribution (OOD) detection (Liu et al., 2020) has aimed to mitigate overconfidence, these methods often fail to reliably flag individual predictions that should be rejected. Misclassification detection (MisD) – also referred to as selective classification or failure prediction – directly addresses this challenge by identifying unreliable predictions for potential rejection or human intervention, thereby reducing the risk of catastrophic failures (Feng et al., 2022).

While MisD is well-established in unimodal settings, with methods spanning confidence-based scoring (Granese et al., 2021; Jiang et al., 2018), outlier exposure (Cheng et al., 2024; Zhu et al., 2023), and confidence learning (Corbière et al., 2019; Moon et al., 2020), its extension to multimodal systems remains largely unexplored. This gap is non-trivial, as unimodal approaches often fail to effectively leverage the complementary information across modalities or to handle failure modes unique to multimodal data, such as signal conflict and misalignment (Rasenberg et al., 2020). Furthermore, some works (Dong et al., 2024; Li et al., 2024a) explore OOD detection with multiple modalities, but their settings fundamentally differ from those of MisD. To illustrate the potential benefits of utilizing multiple modalities for MisD, we present empirical results on the HMDB51 dataset (Kuehne et al., 2011). All models in this analysis were trained solely with a standard cross-entropy loss. As shown in Figure 1 (left), a simple fusion of video and optical flow inputs substantially improves MisD performance – measured by AURC, AUROC, and FPR95 – over unimodal baselines. This finding highlights *the considerable potential of multimodal signals for improving MisD*. Concurrently, Figure 1 (right) reveals that sophisticated OOD detection methods like Energy (Liu et al., 2020),

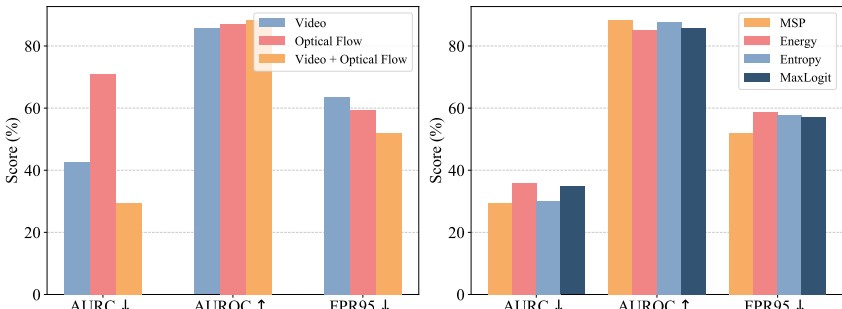

Figure 1: (Left) Multimodal models substantially enhance MisD performance compared to unimodal models, without the need for complex designs. (Right) Advanced OOD detection methods underperform on MisD tasks, while the simple MSP baseline surprisingly remains the most effective.

Entropy (Tian et al., 2022), and MaxLogit (Hendrycks et al., 2019) are outperformed by a simple Maximum Softmax Probability (MSP) baseline (Hendrycks & Gimpel, 2016). Taken together, these findings demonstrate that *merely adapting OOD techniques is insufficient* and motivate the development of dedicated methods tailored for multimodal MisD.

In this work, we identify and systematically characterize the phenomenon of *confidence degradation*, a scenario where the confidence of fused multimodal predictions undesirably falls below that of individual unimodal predictions, particularly in misclassified instances. To address this, we propose MultiMisD, the first dedicated framework for detecting misclassifications in multimodal systems. MultiMisD comprises two key innovations: (1) an *Adaptive Confidence Loss* that explicitly penalizes confidence degradation during training, and (2) *Multimodal Feature Swapping*, a novel augmentation technique that synthesizes challenging, failure-aware training samples by swapping cross-modal embeddings. Training with the confidence penalty and failure-aware outliers improves the model's ability to detect and reject uncertain samples, yielding gains in both accuracy and MisD performance. Comprehensive experiments across five datasets and five modalities demonstrate that MultiMisD sets a new state of the art, outperforming prior best methods by up to $9.58\%$ in AURC, $1.63\%$ in AUROC, and $15.45\%$ in FPR95. Further ablation studies under distribution shifts and multimodal OOD detection settings confirm the robustness and strong generalization of our approach. The primary contributions of this work are:

- We highlight the importance of leveraging multimodal inputs for effective MisD, and provide empirical evidence on the limitations of existing OOD detection approaches in this context.
- We reveal and empirically validate the phenomenon of *confidence degradation* in multimodal models, showing its strong correlation with misclassification.
- We propose MultiMisD, the first dedicated framework tailored to the complex task of multimodal MisD. MultiMisD integrates a novel Adaptive Confidence Loss, addressing the issue of *confidence degradation*, and introduces Multimodal Feature Swapping to further enhance confidence reliability.
- We perform extensive evaluations across diverse datasets and modalities, demonstrating the robustness and effectiveness of MultiMisD in a wide range of scenarios.

## 2 METHODOLOGY

### 2.1 PROBLEM SETUP

**Multimodal Misclassification Detection** aims to detect misclassified samples using ***multiple modalities***. We consider a training set $\mathbb{D} = \{(\mathbf{x}_i, y_i)\}_{i=1}^n$ drawn *i.i.d.* from the joint data distribution $P_{\mathcal{X}\mathcal{Y}}$, where $\mathcal{X}$ is the input space and $\mathcal{Y} = \{1, 2, ..., C\}$ is the label space. Each sample $\mathbf{x}_i$ is composed of $M$ modalities, denoted as $\mathbf{x}_i = \{x_i^k \mid k = 1, \cdots, M\}$. Let $f : \mathcal{X} \mapsto \mathbb{R}^C$ be a neural network trained on samples in $P_{\mathcal{X}\mathcal{Y}}$ that predicts the label of each input sample. The $f$ in multimodal misclassification detection comprises $M$ feature extractors $g_k(\cdot)$ and a classifier $h(\cdot)$. Each feature extractor $g_k(\cdot)$ extracts an embedding $\mathbf{E}^k$ for its corresponding modality $k$, and the classifier $h(\cdot)$ takes the combined embeddings from all modalities as input and outputs a prediction probability $\hat{p}$:

$$\hat{p} = \delta(f(\mathbf{x})) = \delta(h([g_1(x^1), ..., g_M(x^M)])) = \delta(h([\mathbf{E}^1, ..., \mathbf{E}^M])), \tag{1}$$

where $\delta(\cdot)$ is the softmax function. We further include a classifier $h_k(\cdot)$ for each modality $k$ to get predictions from each modality separately, with the prediction probability from the $k$-th modality as $\hat{p}^k = \delta(h_k(g_k(x^k)))$.

To safely deploy classifier $f$ in real-world applications, it should not only be able to make accurate predictions but also distinguish and reject incorrect ones. Formally, let $\kappa(\cdot)$ be a confidence-scoring function that quantifies the model's confidence in its prediction. With a predefined threshold $\tau \in \mathbb{R}^+$, the misclassified samples can be detected based on a decision function $G$ such that for a given input $\mathbf{x}$:

$$G(\mathbf{x}) = \begin{cases} \text{correct} & \text{if } \kappa(\mathbf{x}) \geq \tau, \\ \text{misclassified} & \text{otherwise.} \end{cases} \tag{2}$$

For example, we can easily use MSP (Hendrycks & Gimpel, 2016) as the confidence-scoring function for a given input $\mathbf{x}$ as $\kappa(\mathbf{x}) = \max_{y \in \mathcal{Y}} \hat{p}$. Similarly, other confidence-scoring functions can be adapted from the OOD detection literature, such as MaxLogit (Hendrycks et al., 2019), Energy (Liu et al., 2020), and Entropy (Chan et al., 2021).

## 2.2 Confidence Degradation: A Failure Indicator in Multimodal Systems

We begin by investigating the relationship between multimodal and unimodal prediction confidences to identify systematic patterns that distinguish correct classifications from errors. Our analysis, which uses MSP for confidence scoring, spans four diverse action recognition datasets: HMDB51 (Kuehne et al., 2011), EPIC-Kitchens (Damen et al., 2018), HAC (Dong et al., 2023), and Kinetics-600 (Kay et al., 2017). We consistently observe a specific failure pattern where the confidence of multimodal prediction $\hat{p}$ falls below that of an individual modality $\hat{p}^k$. We formalize this phenomenon as follows:

**Definition 1** (Confidence Degradation). *A sample is considered to exhibit confidence degradation if the confidence of the fused multimodal prediction is strictly lower than that of at least one of its unimodal counterparts:*

$$\exists k \in \{1, \ldots, M\} \quad s.t. \quad \max_{y \in \mathcal{Y}} \hat{p} < \max_{y \in \mathcal{Y}} \hat{p}^k.$$

Figure 2 illustrates the central finding: *confidence degradation is strongly associated with misclassification*. Across all datasets, misclassified samples consistently exhibit a markedly higher rate of degradation than correct predictions, with increases of 32.4% on HMDB51, 23.1% on EPIC-Kitchens, 52.4% on HAC, and 22.0% on Kinetics-600. This suggests that ***failures in multimodal systems frequently coincide with such confidence degradation***. One explanation is that misclassified samples often contain conflicting or ambiguous signals across modalities, which increases uncertainty. When their unimodal outputs are fused, this uncertainty frequently causes the combined confidence to drop

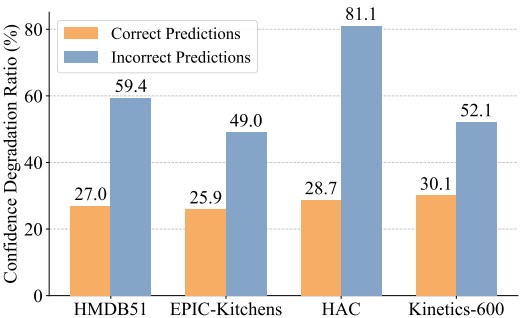

Figure 2: Misclassified samples exhibit a significantly higher proportion of confidence degradation compared to correctly classified ones.

below that of at least one unimodal branch. In contrast, correctly classified samples typically exhibit agreement across modalities, leading to boosted or at least non-degraded fusion confidence. This directly motivates our adaptive training objective, which explicitly penalizes confidence degradation.

## 2.3 Proposed MultiMisD Framework

We introduce MultiMisD, a novel framework for multimodal misclassification detection that integrates two complementary components (Figure 3). First, motivated by the strong correlation between misclassification and the confidence degradation phenomenon, we propose an Adaptive Confidence Loss that directly penalizes this degradation during training. Second, we introduce Multimodal Feature Swapping, an outlier synthesis technique that generates challenging, failure-aware training samples by exchanging cross-modal embeddings. By training on these synthesized failures, MultiMisD learns a more robust uncertainty representation, improving its ability to reject unreliable predictions.

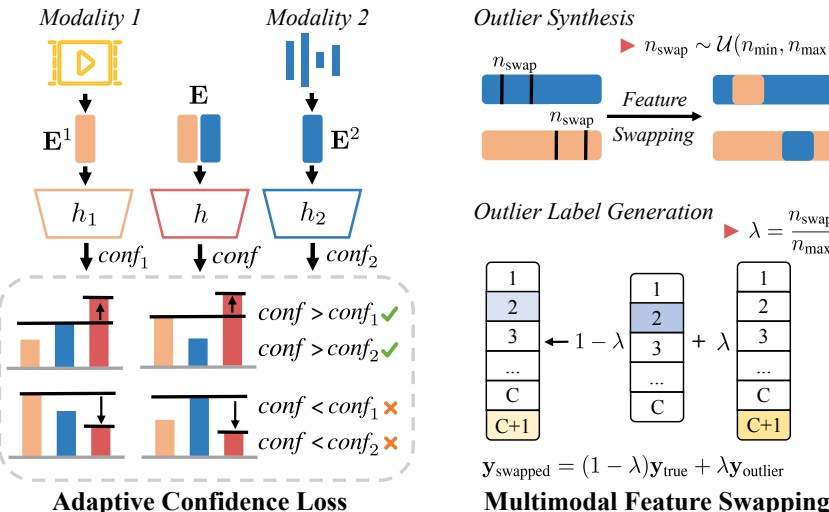

Figure 3: Our MultiMisD framework integrates two principal components. The Adaptive Confidence Loss is designed to penalize the phenomenon of confidence degradation. The Multimodal Feature Swapping serves to generate challenging, failure-aware training instances. This process enables the model to learn to more effectively identify and reject uncertain samples.

The MultiMisD architecture processes inputs from multiple modalities. Each input is passed through a modality-specific encoder to yield an embedding, e.g., $\mathbf{E}^1$ and $\mathbf{E}^2$ for modalities 1 and 2. These embeddings are then concatenated, $\mathbf{E} = [\mathbf{E}^1, \mathbf{E}^2]$, and fed into a fusion classifier to produce the final multimodal prediction $\hat{p}$ with confidence $conf = \max_{y \in \mathcal{Y}} \hat{p}$. In parallel, each unimodal embedding $\mathbf{E}^k$ is also passed through a dedicated classifier to obtain the unimodal prediction $\hat{p}^k$ and its corresponding confidence $conf_k$.

## 2.4 ADAPTIVE CONFIDENCE LOSS

Ideally, effective multimodal fusion should achieve synergy, where the confidence of a fused prediction surpasses that of any single modality, assuming all modalities provide predictive information for the target (Wu et al., 2022). This reflects the successful integration of complementary information to reduce uncertainty and reinforce the decision. However, as we observe in Section 2.2, misclassifications are strongly correlated with confidence degradation, a phenomenon where the fused confidence falls below that of a unimodal counterpart. Such degradation often arises from conflicting or unreliable signals and serves as a strong indicator of prediction failure. Motivated by this observation, we introduce the *Adaptive Confidence Loss* (ACL), which encourages the fused confidence to be at least as high as that of any individual modality. For a two-modality case, ACL is defined as:

$$\mathcal{L}_{\text{acl}} = \frac{1}{2} \left( \max(0, conf_1 - conf) + \max(0, conf_2 - conf) \right). \tag{3}$$

The ACL imposes no penalty when the fused confidence surpasses both unimodal confidences; however, it increasingly penalizes instances where the fused confidence is lower than that of either individual modality. Consequently, ACL encourages the fusion mechanism to learn improved information integration, such that combined evidence from different modalities leads to a more confident prediction. By effectively integrating complementary information from different modalities, *ACL enhances prediction reliability*. Furthermore, *ACL mitigates unimodal overconfidence* by penalizing the model when a high-confidence prediction from one modality conflicts with another. To minimize this cross-modal penalty during training, the model learns to reduce the confidence of the unreliable unimodal stream itself. This process effectively regularizes the unimodal networks, forcing them to become better calibrated and less prone to being "confidently wrong". As a result, the model can integrate information more effectively and produce more reliable multimodal predictions. Additional discussion on ACL is provided in Section G.

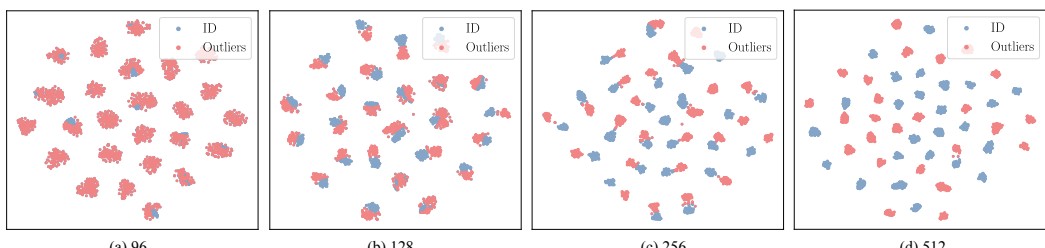

(a) 96         (b) 128         (c) 256         (d) 512

Figure 4: Visualization on outliers generated by Multimodal Feature Swapping with different $n_{\text{swap}}$ (96, 128, 256, 512). Small swaps produce hard negatives that lie near the in-distribution manifold, while larger swaps create more distinct outliers further away.

## 2.5 MULTIMODAL FEATURE SWAPPING

While Outlier Exposure (OE) is an effective technique for improving OOD detection (Hendrycks et al., 2018; Zhang et al., 2023a), it has been shown to be ineffective for MisD (Zhu et al., 2023). This is because OE regularizes the decision boundary by compressing the confidence distribution of in-distribution (ID) samples, which inadvertently makes it harder to distinguish correct ID predictions from incorrect ones. A related challenge, particularly in multimodal settings, is the lack of training data that realistically emulates system failures, such as conflicting modality cues or sensor corruption. Although approaches like OpenMix (Zhu et al., 2023) attempt to address these issues by interpolating between ID and outlier data, they have two critical shortcomings for multimodal tasks. First, they depend on large, auxiliary outlier datasets that are often impractical or unavailable. Second, as a fundamentally unimodal method, OpenMix cannot synthesize the complex failure modes that arise from cross-modal interactions.

To generate challenging, failure-aware outliers without external data, we propose *Multimodal Feature Swapping* (MFS). MFS operates by dynamically swapping multimodal feature embeddings and assigning them corresponding soft labels (as illustrated in Figure 3). By generating outliers directly in feature space, MFS ensures computational efficiency and compatibility with various modalities. MFS is designed to ensure that the synthesized features remain distinct from ID features while preserving semantic consistency. Given ID features $\mathbf{E} = [\mathbf{E}^1, \mathbf{E}^2]$, where $\mathbf{E}^1$ represents features from modality 1 and $\mathbf{E}^2$ from modality 2, MFS randomly selects a subset of $n_{\text{swap}} \sim \mathcal{U}(n_{\text{min}}, n_{\text{max}})$ continuous feature dimensions from each modality. These selected dimensions are then swapped to obtain new feature representations $\widetilde{\mathbf{E}}^1$ and $\widetilde{\mathbf{E}}^2$, which are subsequently concatenated to form the multimodal outlier features $\mathbf{E}_o = [\widetilde{\mathbf{E}}^1, \widetilde{\mathbf{E}}^2]$. A prediction $\hat{p}_o$ is then obtained from $\mathbf{E}_o$ as $\hat{p}_o = \delta(h([\mathbf{E}_o]))$. To supervise these synthesized outliers, we generate soft labels by interpolating between the original ground-truth one-hot label $\mathbf{y}_{\text{true}}$ and an additional class designated for outliers (e.g., $\mathbf{y}_{\text{outlier}} = C + 1$). The weight $\lambda$ for this label interpolation reflects the proportion of features swapped:

$$\mathbf{y}_{\text{swapped}} = (1 - \lambda)\mathbf{y}_{\text{true}} + \lambda\mathbf{y}_{\text{outlier}}, \quad \text{where} \quad \lambda = \frac{n_{\text{swap}}}{n_{\text{max}}}. \tag{4}$$

MFS generates failure-aware outliers by partially swapping cross-modal features. Such swapping preserves intra-modality semantics while disrupting cross-modal consistency—capturing *a critical and common failure mode* in multimodal systems. Figure 4 illustrates a t-SNE visualization of the embedding space under different $n_{\text{swap}}$ values. For small $n_{\text{swap}}$, the generated outliers (red) lie close to the ID clusters (blue), acting as hard negatives. As $n_{\text{swap}}$ increases, the outliers gradually move farther from the ID manifold, confirming that MFS provides a controllable mechanism for generating diverse and realistic failure cases. This property is particularly valuable for training models that must remain sensitive to subtle misclassification signals, especially in multimodal scenarios where errors often stem from partial or conflicting evidence. By introducing corrupted or ambiguous multimodal outliers, MFS encodes the prior knowledge of **what is uncertain and should be assigned low confidence**, thereby teaching the model to recognize broader patterns of uncertainty and enhancing its robustness in detecting real-world misclassifications. Additional discussion on MFS is provided in Section G.

Overall, MFS offers a simple, generalizable, and computationally efficient approach to simulating realistic failure cases for multimodal misclassification detection without requiring external data. The

loss for the synthetic outliers is defined as:

$$\mathcal{L}_{\text{outlier}} = \text{CE}(\hat{p}_o, \mathbf{y}_{\text{swapped}}), \tag{5}$$

where CE denotes the cross-entropy loss. The final training objective integrates all components:

$$\mathcal{L}_{\text{total}} = \mathcal{L}_{\text{cls}} + \mathcal{L}_{\text{outlier}} + \lambda_{\text{acl}}\mathcal{L}_{\text{acl}}, \tag{6}$$

where $\mathcal{L}_{\text{cls}}$ is the cross-entropy loss for the original training samples, and $\lambda_{\text{acl}}$ is a hyperparameter that balances the influence of $\mathcal{L}_{\text{acl}}$.

### 2.6 INFERENCE

Our method focuses on detecting misclassified samples within known classes. Therefore, during the test phase, evaluation is performed exclusively on the original $C$ classes. Specifically, for a given input $\mathbf{x}$, the predicted label is $\hat{y} = \text{argmax}_{y \in \mathcal{Y}} \hat{p}$, and the corresponding confidence is determined using the common MSP score, *i.e.*, $\kappa(\mathbf{x}) = \max_{y \in \mathcal{Y}} \hat{p}$.

## 3 EXPERIMENTS

### 3.1 EXPERIMENTAL SETUP

**Datasets.** We evaluate our proposed framework on four action recognition datasets sourced from the MultiOOD benchmark (Dong et al., 2024): HMDB51 (Kuehne et al., 2011), Kinetics-600 (Kay et al., 2017), HAC (Dong et al., 2023), and EPIC-Kitchens (Damen et al., 2018). Each of these datasets incorporates video and optical flow modalities. For the HAC dataset, we also include evaluations utilizing the audio modality. Further details on each dataset are in the Appendix.

**Implementation.** We conduct experiments across three modalities: video, audio, and optical flow. The MMAction2 (Contributors, 2020) toolkit is adopted for all experiments. To encode visual information, we utilize the SlowFast network (Feichtenhofer et al., 2019), initialized with weights pre-trained on the Kinetics-400 dataset (Kay et al., 2017). For the audio encoder, we employ a ResNet-18 architecture (He et al., 2016), with weights initialized from the VGGSound pre-trained checkpoint (Chen et al., 2020). Similarly, the optical flow encoder uses the SlowFast network, configured with a slow-only pathway and also leveraging pre-trained weights from Kinetics-400 (Kay et al., 2017). The Adam optimizer (Kingma & Ba, 2015) is used for model training, with a learning rate of 0.0001 and a batch size of 16. The hyperparameters for our proposed method are set as follows: $\lambda_{\text{acl}} = 2.0$, $n_{\text{min}} = 32$, $n_{\text{max}} = 256$. We train the models for 50 epochs on an NVIDIA RTX 3090 GPU and select the model with the best performance on the validation dataset.

**Baselines.** We compare our approach against several standard confidence-scoring functions, including MSP (Hendrycks & Gimpel, 2016), MaxLogit (Hendrycks et al., 2019), Energy (Liu et al., 2020), and Entropy (Chan et al., 2021). Additionally, we adapt unimodal MisD methods for our framework, including DOCTOR (Granese et al., 2021) and OpenMix (Zhu et al., 2023), along with the outlier synthesis techniques Mixup (Zhang et al., 2017), RegMixup (Pinto et al., 2022). We also include established training strategies, namely CRL (Moon et al., 2020) and A2D (Dong et al., 2024), where A2D is designed for multimodal OOD detection. These baselines collectively represent a diverse array of techniques for MisD.

**Evaluation Metrics.** Following (Zhu et al., 2023), we assess MisD performance using the following metrics: (1) **AURC** (Area Under the Risk-Coverage Curve), which measures the model's risk (error rate) as a function of coverage (fraction of samples retained). The AURC value is multiplied by $10^3$ following (Zhu et al., 2023). (2) **AUROC** (Area Under the Receiver Operating Characteristic Curve), quantifying the trade-off between the true positive rate (TPR) and the false positive rate (FPR); (3) **FPR95** (False Positive Rate at 95% TPR), indicating the proportion of incorrectly classified samples that are misidentified as correct when the TPR is fixed at $95\%$; (4) **ACC** (Accuracy), representing the standard test accuracy on the ID data.

### 3.2 MAIN RESULTS

**Performance on Multimodal MisD.** Table 1 presents a comparative analysis of our method against various baseline approaches across four datasets—HMDB51, EPIC-Kitchens, HAC, and Kinetics-

| | HMDB51 | | | | EPIC-Kitchens | | | | HAC | | | | Kinetics-600 | | | |
|---|---|---|---|---|---|---|---|---|---|---|---|---|---|---|---|---|
| | AURC↓ | AUROC↑ | FPR95↓ | ACC↑ | AURC↓ | AUROC↑ | FPR95↓ | ACC↑ | AURC↓ | AUROC↑ | FPR95↓ | ACC↑ | AURC↓ | AUROC↑ | FPR95↓ | ACC↑ |
| MaxLogit | 34.76 | 85.65 | 57.02 | 86.20 | 114.22 | 76.92 | 80.00 | 74.25 | 53.61 | 85.12 | 58.97 | 82.11 | 63.90 | 81.25 | 69.85 | 81.24 |
| Energy | 35.78 | 85.03 | 58.68 | 86.20 | 114.91 | 76.67 | 78.95 | 74.25 | 54.33 | 84.80 | 58.97 | 82.11 | 66.48 | 80.12 | 76.58 | 81.24 |
| Entropy | 30.24 | 87.87 | 57.85 | 86.20 | 114.09 | 76.83 | 78.42 | 74.25 | 42.63 | 89.46 | 61.54 | 82.11 | 47.30 | 86.85 | 65.22 | 81.24 |
| MSP | 29.56 | 88.28 | 52.07 | 86.20 | 115.03 | 76.52 | 76.84 | 74.25 | 42.90 | 89.27 | 66.67 | 82.11 | 46.29 | 87.33 | 61.29 | 81.24 |
| DOCTOR | 29.65 | 88.42 | 52.46 | 86.20 | 114.92 | 76.57 | 74.25 | 74.25 | 42.60 | 89.46 | 64.10 | 82.11 | 46.37 | 87.28 | 62.27 | 81.24 |
| Mixup | 36.52 | 87.98 | 50.00 | 84.72 | 110.54 | 77.72 | 75.41 | 75.20 | 34.06 | 87.88 | 55.88 | 84.40 | 50.56 | 86.87 | 60.57 | 80.58 |
| RegMixup | 29.86 | 88.25 | 55.37 | 86.20 | 105.25 | 79.26 | 78.19 | 74.53 | 50.28 | 82.83 | 72.22 | 83.49 | 51.44 | 86.06 | 62.71 | 81.16 |
| OpenMix | 24.15 | 90.13 | 51.33 | 87.12 | 112.14 | 78.46 | 73.68 | 74.25 | 35.42 | 87.51 | 54.84 | 83.03 | 46.73 | 87.69 | 60.27 | 80.79 |
| A2D | 25.01 | 89.79 | 47.01 | 86.66 | 109.90 | 77.85 | 76.72 | 74.39 | 45.89 | 89.77 | 57.14 | 83.94 | 49.26 | 87.97 | 59.79 | 79.97 |
| CRL | 26.44 | 90.39 | 46.40 | 85.75 | 107.42 | 78.33 | 79.17 | 73.98 | 36.46 | 86.53 | 59.38 | 83.49 | 49.16 | 87.29 | 61.73 | 80.47 |
| MultiMisD | **19.97** | **92.02** | **41.96** | **87.23** | **103.25** | **79.27** | **71.58** | **75.20** | **27.41** | **91.48** | **39.39** | **84.86** | **41.85** | **88.99** | **55.89** | **81.45** |

Table 1: MisD performance on action recognition datasets with video and optical flow modalities.

| | video+audio | | | | optical flow+audio | | | | video+optical flow+audio | | | | Average | | | |
|---|---|---|---|---|---|---|---|---|---|---|---|---|---|---|---|---|
| | AURC↓ | AUROC↑ | FPR95↓ | ACC↑ | AURC↓ | AUROC↑ | FPR95↓ | ACC↑ | AURC↓ | AUROC↑ | FPR95↓ | ACC↑ | AURC↓ | AUROC↑ | FPR95↓ | ACC↑ |
| MaxLogit | 28.35 | 84.41 | 68.00 | 88.07 | 117.73 | 83.08 | 72.46 | 68.35 | 26.34 | 87.82 | 75.00 | 87.16 | 57.47 | 85.10 | 71.82 | 81.19 |
| Energy | 29.32 | 83.90 | 68.00 | 88.07 | 120.43 | 82.37 | 75.36 | 68.35 | 27.63 | 86.99 | 75.00 | 87.16 | 59.13 | 84.42 | 72.79 | 81.19 |
| Entropy | 25.10 | 86.03 | 64.00 | 88.07 | 116.95 | 82.75 | 66.67 | 68.35 | 21.40 | 91.00 | 57.14 | 87.16 | 54.48 | 86.59 | 62.60 | 81.19 |
| MSP | 26.79 | 86.30 | 57.69 | 88.07 | 118.04 | 82.46 | 71.01 | 68.35 | 20.99 | 91.35 | 42.86 | 87.16 | 55.27 | 86.70 | 57.19 | 81.19 |
| DOCTOR | 27.10 | 86.02 | 53.85 | 88.07 | 118.03 | 82.51 | 72.46 | 68.35 | 21.03 | 91.33 | 46.43 | 87.16 | 55.39 | 86.62 | 57.58 | 81.19 |
| Mixup | 26.03 | 88.72 | 54.84 | 85.78 | 147.96 | 76.69 | 78.57 | 67.89 | 21.76 | 86.96 | 69.57 | 89.45 | 65.25 | 84.12 | 67.66 | 81.04 |
| RegMixup | 25.58 | 89.56 | 73.33 | 86.24 | 134.26 | 78.00 | 86.96 | 68.35 | 29.77 | 86.43 | 68.97 | 86.71 | 63.20 | 84.66 | 76.42 | 80.43 |
| OpenMix | 26.09 | 88.33 | 64.29 | 87.16 | 117.48 | 81.99 | 71.88 | 70.64 | 20.10 | 91.04 | 48.15 | 87.61 | 54.56 | 87.12 | 61.44 | 81.80 |
| A2D | 59.65 | 81.36 | 60.53 | 82.57 | 108.07 | 81.27 | 79.37 | 71.10 | 20.16 | 90.81 | 48.28 | 86.70 | 62.63 | 84.48 | 62.73 | 80.12 |
| CRL | 26.88 | 89.29 | 43.33 | 86.24 | 118.65 | 83.47 | 71.83 | 67.43 | 31.16 | 87.42 | 64.52 | 85.78 | 58.90 | 86.73 | 59.89 | 79.82 |
| MultiMisD | **24.70** | **89.67** | **41.38** | **86.70** | **98.49** | **83.96** | **63.47** | **71.10** | **15.09** | **92.26** | **34.78** | **89.45** | **46.09** | **88.63** | **46.54** | **82.42** |

Table 2: MisD performance on HAC dataset with different modality combinations.

600—utilizing video and optical flow as input modalities. Our proposed method consistently outperforms all baselines across key MisD metrics. For instance, on the HMDB51 dataset, our method reduces the FPR95 from $52.07\%$ to $41.96\%$ and improves the AUROC from $88.28\%$ to $92.02\%$ when compared to the MSP baseline. On the HAC dataset, it achieves a reduction in AURC from $42.90\%$ to $27.41\%$ and an increase in AUROC from $89.27\%$ to $91.48\%$. Furthermore, on the Kinetics-600 dataset, our method reduces the FPR95 from $61.29\%$ to $55.89\%$. In addition to these MisD improvements, our method even improves classification accuracy on all evaluated datasets. These consistent gains observed across diverse video domains underscore the strong generalization capabilities and robustness of the proposed method.

**Performance under Different Modality Combinations.** Table 2 presents results comparing our method against baseline approaches on the HAC dataset under various modality combinations: video+audio, optical flow+audio, and video+optical flow+audio. While Table 1 exclusively reports results using video and optical flow, this evaluation investigates the generalization of our method across different modality configurations. Our method surpasses baselines in most scenarios, achieving average improvements of $8.39\%$ in AURC, $1.51\%$ in AUROC, and $10.65\%$ in FPR95 relative to the strongest baseline. Concurrently, it enhances classification accuracy from $81.19\%$ to $82.42\%$. These improvements demonstrate the robustness of our approach to diverse modality configurations and its efficacy in enhancing both accuracy and MisD performance.

**Performances under Distribution Shifts.**
In practical applications, environmental conditions can change rapidly (e.g., weather transitioning from sunny to cloudy, then to rainy), necessitating reliable model decisions under such distribution or domain shifts. To simulate these scenarios, we evaluate model performance under various data corruptions with a severity level of 5, including Defocus Blur, Frost, Brightness, Pixelate, and JPEG Compression. Models were trained on the clean HAC dataset using video and optical flow modalities, with corruptions introduced to the video modality exclusively during the testing phase. As illustrated in Figure 5, our framework demonstrates significantly improved MisD performance on AURC under various corruptions in the majority of tested cases.

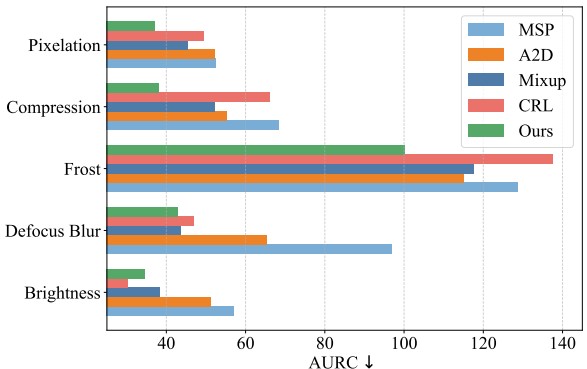

Figure 5: MisD under distribution shift on HAC dataset. The performance of five types of corruption on videos under the severity level of 5 is reported.

| Methods | OOD Datasets | | | | | | | | | | ID ACC ↑ |
|---|---|---|---|---|---|---|---|---|---|---|---|
| | Kinetics-600 | | UCF101 | | EPIC-Kitchens | | HAC | | Average | | |
| | FPR95↓ | AUROC↑ | FPR95↓ | AUROC↑ | FPR95↓ | AUROC↑ | FPR95↓ | AUROC↑ | FPR95↓ | AUROC↑ | |
| MSP | 39.11 | 88.78 | 46.64 | 86.40 | 17.33 | 95.99 | 39.91 | 89.10 | 35.75 | 90.07 | 87.23 |
| +AN | 29.42 | 90.73 | 40.02 | 88.08 | 13.34 | 96.43 | 28.16 | 91.63 | 27.74 | 91.72 | 86.89 |
| +Ours | 27.71 | 92.81 | 34.89 | 89.10 | 12.20 | 97.76 | 21.78 | 94.25 | **24.15** | **93.48** | 87.23 |
| Energy | 32.95 | 92.48 | 44.93 | 87.95 | 8.10 | 97.70 | 32.95 | 92.28 | 29.73 | 92.60 | 87.23 |
| +AN | 24.52 | 93.96 | 36.49 | 89.67 | 6.96 | 97.53 | 22.92 | 94.41 | 22.72 | 93.89 | 86.89 |
| +Ours | 18.59 | 95.39 | 31.58 | 91.05 | 8.32 | 96.57 | 13.45 | 96.07 | **17.99** | **94.77** | 87.23 |
| MaxLogit | 33.07 | 92.31 | 44.93 | 88.02 | 9.12 | 97.77 | 33.06 | 92.17 | 30.05 | 92.57 | 87.23 |
| +AN | 24.86 | 93.69 | 36.60 | 89.71 | 6.96 | 97.67 | 22.92 | 94.22 | 22.84 | 93.82 | 86.89 |
| +Ours | 19.16 | 95.27 | 31.93 | 91.00 | 8.21 | 97.75 | 14.82 | 96.15 | **18.53** | **95.04** | 87.23 |
| GEN | 41.51 | 90.34 | 46.18 | 87.91 | 8.21 | 98.26 | 38.31 | 91.28 | 33.55 | 91.95 | 87.23 |
| +AN | 25.66 | 93.50 | 37.40 | 91.19 | 5.25 | 98.98 | 24.63 | 94.28 | 23.24 | 94.49 | 86.89 |
| +Ours | 22.46 | 95.17 | 31.58 | 92.19 | 2.62 | 99.38 | 15.17 | 96.62 | **17.96** | **95.84** | 87.23 |

Table 5: Multimodal OOD detection using video and optical flow, with HMDB51 as ID.

## 3.3 ABLATION STUDIES

**Effect of Each Component.** Table 3 summarizes performance gains contributed by each component of our framework, evaluated on the HMDB dataset. Commencing from the MSP baseline, the individual incorporation of either the Adaptive Confidence Loss or the Multimodal Feature Swapping module leads to performance improvements across all metrics. Crucially, the combination of both components yields the most substantial overall enhancements. These findings underscore the complementary strengths of the two proposed modules.

| | AURC↓ | AUROC↑ | FPR95↓ | ACC↑ |
|---|---|---|---|---|
| MSP | 29.56 | 88.28 | 52.07 | 86.20 |
| ACL | 24.48 | 90.32 | 43.97 | 86.77 |
| MFS | 25.11 | 90.55 | 46.22 | 86.43 |
| ACL + MFS | **19.97** | **92.02** | **41.96** | **87.23** |

Table 3: Effect of each component.

**Robustness to Different Architectures.** To demonstrate the scalability of our approach, we evaluate its performance using alternative backbone encoders, specifically, I3D (Carreira & Zisserman, 2017) and TSN (Wang et al., 2016), for extracting video and optical flow features. The results on the HMDB51 dataset are presented in Table 4. Despite the utilization of lighter and structurally distinct architectures, our method maintains competitive performance across all four evaluation metrics. It consistently achieves lower FPR95 and AURC, and higher AUROC values compared to all baseline methods.

| | AURC↓ | AUROC↑ | FPR95↓ | Accuracy↑ |
|---|---|---|---|---|
| MSP | 31.77 | 88.17 | 58.59 | 85.40 |
| DOCTOR | 31.76 | 88.18 | 59.38 | 85.40 |
| Mixup | 41.12 | 87.29 | 56.76 | 83.12 |
| RegMixup | 41.67 | 89.41 | 54.27 | 81.30 |
| OpenMix | 33.16 | 88.30 | 55.12 | 84.26 |
| A2D | 34.47 | 88.38 | 55.80 | 84.72 |
| CRL | 37.13 | 89.09 | 60.78 | 82.55 |
| MultiMisD | **27.73** | **90.00** | **51.56** | **85.40** |

Table 4: Ablation on different architectures.

**MultiMisD Improves Multimodal OOD Detection.** A reliable multimodal system should be capable of distinguishing both OOD samples and misclassified ID samples from correct predictions. Therefore, in addition to MisD, we investigate the OOD detection capabilities of our method using the MultiOOD benchmark (Dong et al., 2024), focusing on video and optical flow modalities. The ID dataset employed is HMDB51. For OOD datasets, we utilize Kinetics-600, UCF101 (Soomro et al., 2012), EPIC-Kitchens, and HAC. Performance is evaluated using AUROC, FPR95, and ID Accuracy. We train models using both the A2D+NP-Mix (AN) (Dong et al., 2024) strategies in MultiOOD and our MultiMisD framework, and subsequently evaluate them with various OOD confidence-scoring functions, including MSP, Energy, MaxLogit, and GEN (Liu et al., 2023). The results, presented in Table 5, indicate that MultiMisD not only exhibits strong MisD capabilities but also achieves robust OOD detection performance in comparison to AN.

**MisD in Presence of OOD Samples.** In this challenging setting, OOD samples are introduced into the test data, and the model is required to distinguish both OOD samples and misclassified ID samples from correct predictions. We use HMDB51 as the ID dataset and incorporate OOD samples from the HAC dataset. As shown in Table 6, our MultiMisD demonstrates strong robustness in this scenario, accurately identifying both OOD and misclassified samples. It achieves average improvements of $1.47\%$ in AUROC, $4.45\%$ in FPR95, and $0.57\%$ in ACC compared to the strongest baseline.

| | AUROC↑ | FPR95↓ | ACC↑ |
|---|---|---|---|
| MSP | 94.14 | 39.35 | 86.20 |
| Mixup | 95.35 | 24.92 | 84.72 |
| A2D | 94.33 | 29.90 | 86.66 |
| CRL | 94.14 | 32.29 | 85.75 |
| MultiMisD | **96.82** | **20.47** | **87.23** |

Table 6: Performance under both OOD and misclassified samples.

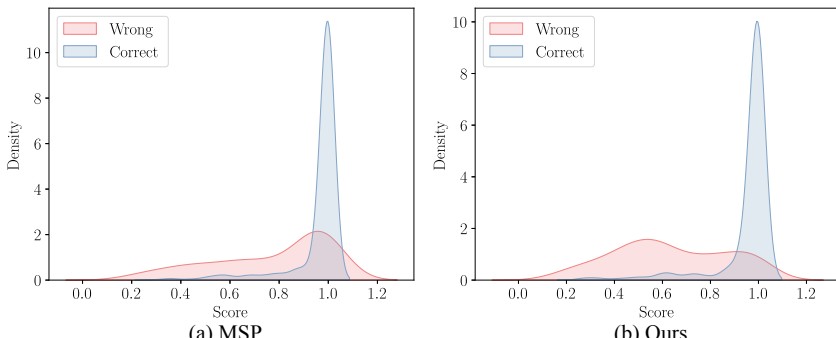

(a) MSP

(b) Ours

Figure 6: Score distribution for correct and wrong predictions. Our method leads to more clearly separated distributions, thereby enhancing the efficacy of misclassification detection.

**Compare with Other Feature-space Augmentation Methods.** To assess the effectiveness of alternative augmentation strategies, we replace MFS with Random Noise (randomly replacing embedding values with noise), Random Drop (randomly replacing embedding values with zeros), and Feature Mixing (Liu et al., 2025). As shown in Table 7, all baseline methods improve MisD performance, highlighting the importance of outlier synthesis for regularization. However, MFS proves most effective, as it generates outliers of varying difficulty that better capture cross-modal inconsistency.

|  | AURC↓ | AUROC↑ | FPR95↓ | ACC↑ |
|---|---|---|---|---|
| MSP | 29.56 | 88.28 | 52.07 | 86.20 |
| Random Noise | 24.86 | 90.82 | 42.86 | 86.43 |
| Random Drop | 22.80 | 91.24 | 46.09 | 86.89 |
| Feature Mixing | 21.79 | 91.33 | 42.11 | 87.00 |
| MFS | **19.97** | **92.02** | **41.96** | **87.23** |

Table 7: Compare with other feature space augmentation methods.

**Evaluation on More Modalities.** To further evaluate the robustness of our framework, we conduct experiments on the SemanticKITTI dataset (Behley et al., 2019), using image and LiDAR point cloud modalities for the 3D semantic segmentation task. We adopt the fusion framework of Zhuang et al. (2021) and adapt the evaluation to the MisD setting. As shown in Table 8, our framework achieves strong misclassification detection performance for the semantic segmentation task with image and LiDAR modalities.

|  | AURC↓ | AUROC↑ | FPR95↓ | mIoU↑ |
|---|---|---|---|---|
| MSP | 33.90 | 79.97 | 55.49 | 59.25 |
| MultiMisD | **21.90** | **84.51** | **52.51** | **63.56** |

Table 8: Results on SemanticKITTI for 3D semantic segmentation task.

**Visualization.** To qualitatively assess confidence score distributions, we visualized them for correct and incorrect predictions on the HMDB51 dataset, as depicted in Figure 6. The baseline MSP method exhibits a less distinct separation in confidence scores between correctly classified and misclassified samples. In contrast, our proposed solution assigns higher confidence scores to correct predictions and discernibly lower scores to incorrect ones. This leads to more clearly separated distributions, thereby enhancing the efficacy of misclassification detection.

## 4 CONCLUSION

In this work, we addressed the critical yet underexplored challenge of misclassification detection in multimodal systems, a vital component for ensuring reliability in safety-sensitive domains. We introduced MultiMisD, the first dedicated framework designed to tackle this problem. By characterizing the confidence degradation phenomenon—where fused multimodal predictions exhibit lower confidence than their unimodal counterparts in most error cases—we introduced an Adaptive Confidence Loss that directly penalizes this behavior during training. Complementing this loss, our Multimodal Feature Swapping technique synthesizes challenging outliers, further enhancing the model's ability to flag unreliable predictions. Extensive evaluations across four diverse datasets and three modalities demonstrated MultiMisD's superior performance and generalization capabilities, significantly outperforming existing baselines. By enabling more reliable identification of untrustworthy predictions, MultiMisD represents a significant step towards enhancing the safety and trustworthiness of multimodal systems in real-world applications.

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

## A  THE USE OF LARGE LANGUAGE MODELS

Large language models (LLMs) were used solely as assistive tools for improving the readability of this paper. In particular, they were employed to polish the writing style, correct grammar, and fix spelling mistakes. All research ideas, experiments, analyses, and conclusions were conceived and conducted entirely by the authors without assistance from LLMs.

## B  BROADER IMPACT, LIMITATIONS, AND FUTURE WORK

**Broader Impact.** The MultiMisD framework presented in this work has the potential to generate a significant positive societal impact. As AI systems, especially multimodal ones, are increasingly integrated into safety-critical applications such as autonomous driving, medical diagnosis, and industrial robotics, the ability to reliably detect potential misclassifications is paramount. MultiMisD directly contributes to enhancing the safety and trustworthiness of these systems by providing a dedicated mechanism to identify and flag uncertain predictions before they can lead to adverse outcomes. This can foster greater public trust and accelerate the responsible adoption of AI technologies in areas where reliability is non-negotiable. Furthermore, by improving the understanding of failure modes in multimodal models, such as the identified confidence degradation phenomenon, our work can guide the development of more robust and dependable AI systems across various domains, ultimately leading to safer and more effective human-AI collaboration.

**Limitations.** While MultiMisD demonstrates strong gains across multiple datasets and modalities, it has several limitations. First, while we demonstrate robustness under certain distribution shifts and OOD scenarios, the behavior of MultiMisD against sophisticated adversarial attacks specifically designed to fool the misclassification detector itself remains an open area. Second, our current study primarily focuses on specific types of modalities (e.g., video, optical flow, and audio). The generalization of MultiMisD to a very broad range of disparate modalities (e.g., text with medical imaging, or sensor data with acoustic signals) without specific adaptations warrants further investigation.

**Future Work.** Building upon the contributions of this paper, several avenues for future research present themselves. First, we plan to explore the integration of MultiMisD with online learning and continual learning paradigms. This would allow the misclassification detector to adapt dynamically to evolving data distributions and novel failure modes encountered during real-world deployment, which is crucial for long-term operational reliability. Second, extending and evaluating MultiMisD across a wider spectrum of modalities (e.g., language, audio, tabular data) and a more diverse set of complex, safety-critical tasks (e.g., real-time robotic interaction) is a key direction. Finally, investigating the robustness of MultiMisD against targeted adversarial attacks and developing defense mechanisms will be crucial for deployment in adversarial settings.

## C  RELATED WORK

### C.1  MISCLASSIFICATION DETECTION

Foundational work on failure prediction traces back to the concept of selective classification, which formalizes the trade-off between prediction accuracy and abstention when model confidence is low (Chow, 1970). In the deep learning era, this concept has re-emerged as misclassification detection—approaches that enable a model to identify when its output is likely to be incorrect (Qiu & Miikkulainen, 2022). While thresholding the MSP (Hendrycks & Gimpel, 2016) remains a simple yet effective baseline, its susceptibility to overconfidence limits its reliability (Qiu & Miikkulainen, 2022; Pinto et al., 2022). A variety of methods have been proposed to more effectively estimate prediction risk. One stream of research focuses on training auxiliary modules, such as heads or separate networks, to explicitly predict correctness based on intermediate features or model logits (Corbière et al., 2019; Jiang et al., 2018; Granese et al., 2021). For instance, ConfidNet (Corbière et al., 2019) introduces a dedicated confidence estimation head that operates on penultimate-layer features. A distinct line of research adopts an integrated approach, jointly optimizing MisD functionalities with the primary model's training objective. Representative techniques in this category include improving the separability of correct and incorrect representations (Luo et al., 2021), enhancing confidence ranking (Moon et al., 2020), regularizing based on sample typicality (Liu et al., 2024),

and enforcing flatter loss landscapes around misclassified examples (Zhu et al., 2022). Concurrently, data augmentation strategies have proven effective, primarily by exposing the model to synthesized failure cases during training (Zhu et al., 2023; Li et al., 2024b; Cheng et al., 2024; Han & Zhang, 2024). However, all previous approaches were designed for unimodal scenarios, without accounting for the interaction and complementary nature of diverse modalities.

## C.2 OUT-OF-DISTRIBUTION DETECTION

OOD detection shares a similar objective with MisD but addresses fundamentally different challenges. Specifically, OOD detection aims to identify test samples that exhibit semantic shifts from the training distribution, typically without compromising in-distribution (ID) classification accuracy. OOD detection has been extensively investigated in recent years (Yang et al., 2021b; Zhang et al., 2023b), encompassing diverse approaches such as post-hoc scoring (Hendrycks & Gimpel, 2016; Liang et al., 2017; Liu et al., 2020), feature-based techniques (Lee et al., 2018; Sun et al., 2022), outlier exposure (Hendrycks et al., 2018; Yu & Aizawa, 2019; Yang et al., 2021a), and reconstruction-based methods (Di Biase et al., 2021; Zhou, 2022). Multimodal OOD detection (Dong et al., 2024; Li et al., 2024a) is also an emerging research area. Although OOD detection methods are often employed as baselines for MisD, recent studies (Zhu et al., 2022; Jaeger et al., 2022; Zhu et al., 2023) demonstrate that techniques optimized for OOD detection generally exhibit suboptimal performance on MisD tasks. This finding underscores the necessity for developing specialized MisD approaches.

## D FURTHER DETAILS ON DATASETS

Our framework is primarily evaluated on four action recognition datasets from the MultiOOD benchmark (Dong et al., 2024): HMDB51 (Kuehne et al., 2011), EPIC-Kitchens (Damen et al., 2018), HAC (Dong et al., 2023), and Kinetics-600 (Kay et al., 2017). For evaluating multimodal Out-of-Distribution (OOD) detection in ablation studies, we additionally employ the UCF101 dataset (Simonyan & Zisserman, 2014), also from the MultiOOD benchmark.

**HMDB51** is an action recognition dataset comprising 6,766 video clips distributed across 51 action categories. The videos are sourced from digitized movies and YouTube. This dataset includes both video and optical flow modalities. **EPIC-Kitchens** is an egocentric video dataset capturing daily kitchen activities recorded by 32 participants. For our experiments, we utilize a subset of 4,871 video clips from participant P22, encompassing eight common actions (*put*, *take*, *open*, *close*, *wash*, *cut*, *mix*, and *pour*). The dataset provides video and optical flow modalities. **Kinetics-600** is a large-scale action recognition dataset containing approximately 480,000 10-second clips distributed across 600 action classes. Following (Dong et al., 2024), we utilize a subset of 100 classes, resulting in 24,981 video clips for our study. This dataset offers video, audio, and optical flow modalities. **HAC** contains 3,381 video clips featuring seven action categories (e.g., *sleeping*, *watching TV*, *eating*, *running*) performed by humans, animals, and cartoon characters. The dataset includes video, optical flow, and audio modalities. **UCF101** is a diverse video action recognition dataset consisting of 13,320 clips across 101 action classes. The videos, sourced from YouTube, exhibit significant variation in camera motion, object appearance, scale, pose, viewpoint, and background. This dataset provides video and optical flow modalities.

## E MORE IMPLEMENTATION DETAILS

**Pseudo Code for Multimodal Feature Swapping.** We provide the pseudo code for multimodal outlier synthesis in Algorithm 1, where we dynamically swap multimodal feature embeddings and assign them corresponding soft labels.

**Extension to More Modalities.** Our framework is not limited to two modalities and can be easily extended to $M$ modalities. Given a training sample $\mathbf{x}$ with $M$ modalities, we obtain prediction confidence score *conf* from the combined embeddings of all modalities, and $conf_1, conf_2, ..., conf_M$ from each modality. The Adaptive Confidence Loss can then be defined as:

$$\mathcal{L}_{\text{acl}} = \frac{1}{M} \sum_{i=1}^{M} \max(0, conf_i - conf). \tag{7}$$

---

**Algorithm 1** Multimodal Feature Swapping

---

**Input:** ID feature $\mathbf{E} = [\mathbf{E}^1, \mathbf{E}^2]$, where $\mathbf{E}^1$ and $\mathbf{E}^2$ are from modality 1 and 2; minimum and maximum number $n_{\min}$ and $n_{\max}$ for swapping; ground-truth one-hot label $\mathbf{y}_{\text{true}}$ for $\mathbf{E}$ and $\mathbf{y}_{\text{outlier}} = C + 1$.

**Pseudo Code:**

Sample $n_{\text{swap}} \sim \mathcal{U}(n_{\min}, n_{\max})$, $\lambda = \frac{n_{\text{swap}}}{n_{\max}}$;

Randomly select start indices $s_1$, $s_2$ for modality 1 and 2;

Clone features $\widetilde{\mathbf{E}}^1 \leftarrow \mathbf{E}^1$, $\widetilde{\mathbf{E}}^2 \leftarrow \mathbf{E}^2$;

Swap $n_{\text{swap}}$ dimensions across modalities:

$$\widetilde{\mathbf{E}}^1[s_1:s_1+n_{\text{swap}}] \leftarrow \mathbf{E}^2[s_2:s_2+n_{\text{swap}}]$$

$$\widetilde{\mathbf{E}}^2[s_2:s_2+n_{\text{swap}}] \leftarrow \mathbf{E}^1[s_1:s_1+n_{\text{swap}}]$$

Generate label for outlier feature $\mathbf{y}_{\text{swapped}} = (1 - \lambda)\mathbf{y}_{\text{true}} + \lambda\mathbf{y}_{\text{outlier}}$.

**Output:** Multimodal outlier feature $\mathbf{E}_o = [\widetilde{\mathbf{E}}^1, \widetilde{\mathbf{E}}^2]$ and label $\mathbf{y}_{\text{mixed}}$.

---

| | AURC↓ | AUROC↑ | FPR95↓ | ACC↑ |
|---|---|---|---|---|
| 128 | 29.08 | 88.27 | 49.57 | 86.66 |
| 256 | 25.11 | 90.55 | 46.22 | 86.43 |
| 512 | 25.34 | 90.98 | 43.90 | 85.97 |

Table 9: Effect of $n_{max}$ in MFS.

| | AURC↓ | AUROC↑ | FPR95↓ | ACC↑ |
|---|---|---|---|---|
| 0.2 | 24.42 | 90.19 | 46.15 | 86.66 |
| 0.5 | 24.18 | 90.28 | 46.22 | 87.00 |
| 1.0 | 23.11 | 90.93 | 42.11 | 86.55 |
| 2.0 | 19.97 | 92.02 | 41.96 | 87.23 |

Table 10: Effect of weight $\lambda_{\text{acl}}$ for ACL.

## F  MORE ABLATION STUDIES

**Parameter Sensitivity.** We evaluate the sensitivity of our framework to two key hyperparameters using the HMDB51 dataset. First, the maximum swapping dimension $n_{max}$ for Multimodal Feature Swapping (MFS) was varied among 128, 256, and 512, with results presented in Table 9. An $n_{max}$ value of 256 yielded the optimal balance, achieving robust performance across all evaluation metrics. Subsequently, with $n_{max}$ fixed at 256, the weight $\lambda_{\text{acl}}$ for Adaptive Confidence Loss (ACL) was evaluated over the set 0.2, 0.5, 1.0, and 2.0 (detailed in Table 10). A value of $\lambda_{\text{acl}} = 2.0$ consistently delivered the strongest MisD performance. Importantly, the framework's performance remained stable across both parameter sweeps, underscoring its robustness to variations in these hyperparameters.

## G  FURTHER DISCUSSIONS ON PROPOSED MODULES

**How Adaptive Confidence Loss Addresses Unimodal Overconfidence.** Imagine a situation where one modality (e.g., audio) is ambiguous or corrupted, causing its corresponding unimodal network to make a confidently wrong prediction (high $conf_1$). The other modality (e.g., video) provides clear evidence for the correct class, leading to a correct, high-confidence prediction ($conf_2$). A standard fusion model might struggle to integrate these conflicting signals. If the fusion process results in a moderate fused confidence $conf$ that is lower than the erroneously high $conf_1$, it will incur a large ACL penalty from the $\max(0, conf_1 - conf)$ term. To minimize this loss, the model can't easily force the fused confidence up without a good reason, as that would be penalized by the cross-entropy loss term $\mathcal{L}_{\text{cls}}$ if the prediction is wrong. Instead, a more effective way to reduce the ACL loss is to lower the confidence of the overconfident unimodal prediction ($conf_1$). Through repeated exposure to such conflicting examples during training, the unimodal feature extractor learns a valuable lesson. It learns that producing a high-confidence prediction that is likely to be contradicted by another modality is "expensive" in terms of the overall loss. Therefore, it adjusts its weights to become more cautious and better calibrated. The unimodal network learns to associate maximum confidence not just with strong internal features, but with features that are also robust and likely to lead to cross-modal agreement. As shown in Fig. 7 and Fig. 8, training with ACL effectively alleviates unimodal overconfidence on incorrect predictions across all datasets.

**Adaptive Confidence Loss Does Not Induce Overconfidence in Incorrect Predictions.** ACL is designed to encourage the fused confidence to be at least as high as that of any individual modality, but

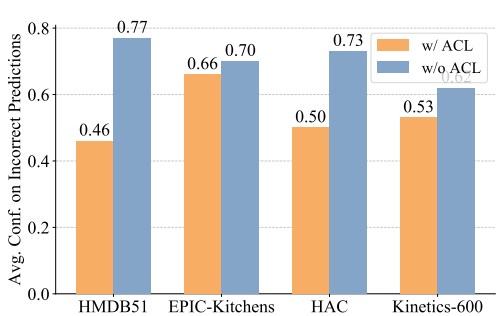

Figure 7: The average confidence on incorrect predictions for video modality w/ and w/o ACL.

Figure 8: The average confidence on incorrect predictions for optical flow modality w/ and w/o ACL.

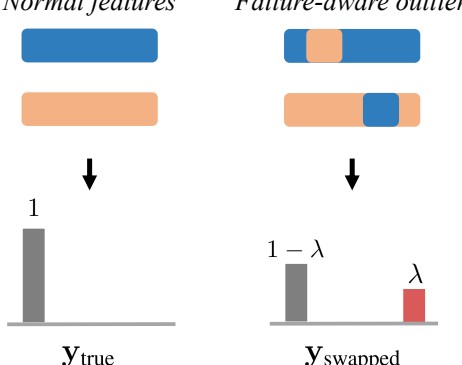

Figure 9: An illustration of how MFS enables the detection and rejection of uncertain predictions, thereby improving model reliability. The failure-aware outliers generated by MFS are treated as uncertain because they introduce cross-modal inconsistency. Accordingly, their soft label $\mathbf{y}_{\text{swapped}}$ reduces the ground-truth class probability from 1 to $1 - \lambda$, compared to $\mathbf{y}_{\text{true}}$. Training with cross-entropy loss on $\mathbf{y}_{\text{swapped}}$ ensures that the prediction confidence on failure-aware outliers remains lower than on normal features, which is exactly the desired outcome.

it does not make fused predictions overconfident when they are wrong. Specifically, when predictions are correct, the cross-entropy loss term $\mathcal{L}_{\text{cls}}$ and ACL act in synergy, both encouraging higher fused confidence. In contrast, when predictions are incorrect, a higher fused confidence increases the cross-entropy loss $\mathcal{L}_{\text{cls}}$, which counteracts the ACL term. This balance ensures that ACL boosts confidence only for correct predictions while avoiding overconfidence in errors.

**Multimodal Feature Swapping.** Our MFS module isn't designed to replicate the entire, complex distribution of all possible real-world failures, as this would be intractable. Instead, MFS serves as a principled and targeted regularizer that exposes the model to a critical and common failure mode in multimodal systems: cross-modal inconsistency. Our core hypothesis is that many real-world errors arise from conflicting or ambiguous signals between modalities. MFS directly simulates this failure mode by swapping feature segments, creating semantically incoherent but challenging "hard negative" samples that are near the in-distribution data. Training on these synthesized samples forces the model to develop a more robust fusion mechanism that must critically assess the semantic agreement between modalities. This closer examination of cross-modal consistency improves its ability to detect real-world misclassifications, which often exhibit similar signal conflicts. The effectiveness of this approach is validated by our extensive empirical results, which demonstrate that learning to reject these synthetic inconsistencies directly translates to better identification of real-world errors. Fig. 9 gives a detailed illustration of how MFS enables the detection and rejection of uncertain predictions, thereby improving model reliability.

