# OpenReview forum: "MultiMisD: Multimodal Misclassification Detection"
_ICLR.cc/2026/Conference — ICLR 2026 Conference Withdrawn Submission_

### Official Review · Reviewer_UrXG · 2025-10-31

**Soundness:** 3
**Presentation:** 3
**Contribution:** 2
**Rating:** 2
**Confidence:** 4

**Summary:**

This paper presents multimodal misclassification detection (MultiMisD). The authors employs an adaptive confidence loss (ACL) to mitigate this degradation and multimodal feature swapping (MFS) to create synthetic failure-aware examples. Experiments on datasets demonstrate that the proposed method achieves state-of-the-art results for multimodal MisD.

**Strengths:**

- The paper addresses an important and safety-critical problem, i.e., detecting failures in multimodal systems, which is a necessary and relevant distinction for improving model reliability.
- The finding of confidence degradation is convincing in common multimodal fusion strategies. The proposed method is effective on several benchmarks.

**Weaknesses:**

- The confidence degradation phenomenon describes that when two confident modalities disagree, the fuser outputs a lower confidence. This is an expected behavior of a model struggling to reconcile conflicting signals. Framing this expected loss of confidence during a failure as a degradation misunderstands the role of the fusion module as a conflict resolver.

- The proposed ACL is a heuristic penalty designed to force the fused prediction confidence to remain high if any unimodal branch is confident. This mechanism is equivalent to actively destroying the calibration of the fused output. By artificially inflating the confidence of the multimodal classifier based on uncalibrated unimodal signals, the method is masking genuine uncertainty. The paper fails to prove that this enforced confidence boost is not simply trading true uncertainty for overconfidence on hard, but correct, samples.
- The entire framework is demonstrated predominantly on simple fusion models. State-of-the-art multimodal systems, especially those using Transformer architectures, already employ sophisticated cross-attention or dynamic gating mechanisms specifically designed to weight and filter out unreliable or conflicting modalities. The paper fails to demonstrate that MultiMisD provides a substantial benefit on top of these modern, inherently robust fusion methods, making the contribution potentially relevant only to outdated baseline architectures.

**Questions:**

- Given that ACL actively encourages high fused confidence, please provide a rigorous calibration analysis (e.g., Expected Calibration Error (ECE)) for the fused model with and without ACL. Is the gain in MisD performance achieved at the unacceptable cost of catastrophic overconfidence on correctly classified but inherently uncertain samples?

---

### Official Review · Reviewer_4WQ2 · 2025-10-31

**Soundness:** 3
**Presentation:** 3
**Contribution:** 3
**Rating:** 6
**Confidence:** 4

**Summary:**

Addressing the misclassification detection (MisD) problem in multimodal models, the authors, based on the discovery of "confidence degradation after multimodal fusion," point out that for misclassified samples, the confidence of the fused multimodal prediction is lower than that of at least one unimodal branch. They propose the MultiMisD framework, which includes two key components: Adaptive Confidence Loss (ACL) and Multimodal Feature Swapping (MFS). ACL is a novel loss function that penalizes the "confidence degradation" phenomenon during training; MFS is used to generate anomaly samples suitable for multimodal methods in the feature space. The framework is evaluated on four different datasets and three modalities.

**Strengths:**

1. The design of ACL is effective and intuitive, directly targeting the "confidence degradation" problem by using a simple loss term to encourage the fused confidence not to be lower than any unimodal confidence.
2. MFS requires no additional data and simulates modal conflicts or inconsistencies by swapping cross-modal information in the feature space, enhancing the model's ability to handle real-world failures.

**Weaknesses:**

1. ACL has a potential risk: if all unimodal branches provide high-confidence but incorrect predictions, the ACL loss will be zero, which may encourage the model to make high-confidence errors.
2. The paper only tests on 2–3 modalities, but for cases with a large number of modalities or extremely different modalities, the method's effectiveness remains to be verified.
3. How is the preset range for the number of swapped dimensions in MFS determined? In cases with significant differences in modal dimensions, how should the impact of the preset range be balanced?

**Questions:**

1. How is the preset range in MFS determined? Is this range universal for all datasets and networks? If not, could adaptive selection be considered?
2. Have you considered testing the scenario where "all unimodals provide high-confidence incorrect predictions" through specific data construction or adversarial examples?
3. Figure 2 shows that even for correct predictions, about 25–30% of samples exhibit "confidence degradation." Does this mean that ACL is penalizing these samples that should be correct? Why does MultiMisD not only not reduce accuracy but actually improve it in this case? Would model performance be better if these samples were ignored during training?

---

### Official Review · Reviewer_GbHv · 2025-11-01

**Soundness:** 2
**Presentation:** 2
**Contribution:** 2
**Rating:** 2
**Confidence:** 4

**Summary:**

This paper proposes MultiMisD to detect prediction failures under multimodal scenarios. The paper first identifies the confidence degradation phenomenon and proposes the ACL loss to penalize the confidence degradation. The authors further proposes to synthesize outliers via feature swapping for more generalizability.

**Strengths:**

1. The proposed MultiMisD is straightforward and intuitive, which penalizes the confidence degradation during the modality fusion process. The empirical results demonstrate that the MultiMisD is effective in almost all cases.

**Weaknesses:**

1. The observation of confidence degredation is not sufficient and not discussed in-depth. The authors claim that the low confidence can be related to failure predictions in multimodal systems, but further theoretical or empirical evidence is missing to support it.
2. Straightforward but lack insight. The author proposes to penalize the confidence degradation. However, the confidence degradation is not well demonstrated to be harmful for multimodal predictions. Therefore, by forcing high confidence for multimodal predictions may lead to potential failures in cases where the model ought to be cautious and produce low confidence when various information from different modalities is considered.
3. The outlier synthesis method is not well grounded. The authors generate outlier features by simply swapping features among different modalities. However, the correctness of the generated features is neither discussed nor guaranteed, as the generated abstract features may interfere with known semantics and thus lead to false outliers.
4. The problem definition is not clear. While the author claims that they proposed MultiMisD to handle the failure prediction, they report OOD detection performance to validate the effectiveness of MultiMisD. This is confusing and makes me wonder what the exact problem that the authors want to handle. To my knowledge, failure prediction deals with false prediction on in-distribution data, and OOD detection deals with out-of-distribution data.

**Questions:**

See the weaknesses above.

---

### Official Review · Reviewer_fs8V · 2025-11-04

**Soundness:** 3
**Presentation:** 4
**Contribution:** 3
**Rating:** 6
**Confidence:** 4

**Summary:**

This paper introduces MultiMisD, a framework for detecting misclassifications in multimodal systems such as video, optical flow, and audio models. The authors start from the observation that in multimodal models, the combined prediction is often less confident than one or more unimodal branches when the model is wrong. They call this confidence degradation and show that it strongly correlates with misclassification. The method adds two new components to the training process. The first is an Adaptive Confidence Loss that penalizes cases where the fused confidence is lower than the unimodal confidences. The second is Multimodal Feature Swapping, which generates failure-aware examples by exchanging parts of embeddings across modalities and using soft labels to train the model to recognize uncertainty. Together these two modules are trained jointly with the classifier. The paper shows large improvements on multiple video and audio datasets in metrics such as AURC, AUROC, and FPR95. The method also generalizes to other modality combinations, improves robustness to distribution shifts, and enhances OOD detection.

**Strengths:**

The paper addresses an important  problem of detecting misclassifications in multimodal models. The motivation is clear and supported by analysis showing the link between confidence degradation and misclassification. The proposed Adaptive Confidence Loss is simple, well-motivated, and relatively easy to add to existing multimodal architectures that are based on fusion of individual extractors. Multimodal Feature Swapping is interesting and does not need any external outlier dataset, making it efficient and general. The experiments are extensive, covering four datasets and several modality combinations, and the improvements are consistent. The ablation studies clearly separate the effect of each component.

**Weaknesses:**

The framework requires retraining the model, so it cannot be applied post hoc to existing multimodal systems. The analysis of why confidence degradation occurs is empirical and does not have a strong theoretical explanation. It is also not clear whether it is a feature or a bug. Given that most deep learning models are poorly calibrated, one would expect that adding more modalities should reduce the chance of confidently being wrong. The authors mention this point later in the ACL discussion, but the motivation seems heavily tied to the observations from Figure 2, which are purely empirical and not generalized enough to other problems. Therefore, related to my earlier point, the method assumes that both the final classifier and the unimodal feature extractors are trainable, which is not practical in many scenarios. At least for ACL, one of the baselines should include simple post-hoc calibration (such as temperature scaling) of the unimodal feature extractors, followed by applying the loss only to the classifier h. This would ensure that the unimodal confidences are calibrated, allowing the loss to directly inform the classifier rather than the current formulation of updating the feature extractors. In fact, a simple reinforcement-style penalty could suffice in such a setup.

Further, the idea of swapping features assumes that the modalities have compatible feature dimensions or that they can be projected to the same space, which may not always be straightforward. Related to this, given the conceptual overlap between the proposed MFS and the feature-swapping idea in https://arxiv.org/pdf/2505.16985v1, this deserves a separate discussion. What exactly is novel about this approach compared to theirs? Liu et al. apply the idea for OOD detection in semantic segmentation, while this paper focuses only on classification tasks. Is the soft-label generation the main novelty difference? The reviewer is aware of Table 7, but given how close the two methods are, this comparison is insufficient, and a more detailed discussion of novelty is needed.

The corruption study uses standard photometric and compression artifacts. However, multimodal systems can fail in several other ways such as missing or noisy modalities and modality conflicts induced by priors. These aspects are not explored or discussed. The paper also presents MFS primarily for two modalities, both in the figures and examples. It is unclear how this extends to a third modality. How is E₃ handled in E = [E₁, E₂] during swapping, and how was MFS implemented for the three-modality results in Table 2 that include audio?

While outlier synthesis is one important direction for failure prediction, there are parallel efforts using interpretability-based approaches such as https://dl.acm.org/doi/10.1007/978-3-031-72986-7_27 and https://openreview.net/forum?id=99RpBVpLiX. These should be discussed as related alternatives. Even within outlier synthesis, another relevant reference is https://proceedings.mlr.press/v227/narayanaswamy24a/narayanaswamy24a.pdf, which should be acknowledged for completeness.

**Questions:**

Please refer to my weaknesses

---

### Note · Authors · 2025-11-14

I have read and agree with the venue's withdrawal policy on behalf of myself and my co-authors.